

# The impact of improved satellite retrievals on estimates of biospheric carbon balance

Scot M. Miller[1] and Anna M. Michalak[2]

[1]Department of Environmental Health and Engineering, Johns Hopkins University, Baltimore, MD, USA
[2]Department of Global Ecology, Carnegie Institution for Science, Stanford, CA, USA

**Correspondence:** Scot M. Miller (smill191@jhu.edu, scot.m.miller@gmail.com)

**Abstract.** The Orbiting Carbon Observatory 2 (OCO-2) is NASA's first satellite dedicated to monitoring $CO_2$ from space and could provide novel insight into $CO_2$ fluxes across the globe. However, one continuing challenge is the development of a robust retrieval algorithm: an estimate of atmospheric $CO_2$ from satellite observations of near infrared radiation. The OCO-2 retrievals have undergone multiple updates since the satellite's launch, and the retrieval algorithm is now on its ninth version. Some of
these retrieval updates, particularly version 8, led to marked changes in the $CO_2$ observations, changes of 0.5 ppm or more. In this study, we evaluate the extent to which current OCO-2 observations can constrain monthly $CO_2$ sources and sinks from the biosphere, and we particularly focus on how this constraint has evolved with improvements to the OCO-2 retrieval algorithm. We find that improvements in the $CO_2$ retrieval are having a potentially transformative effect on satellite-based estimates of the global biospheric carbon balance. The version 7 OCO-2 retrievals formed the basis of early inverse modeling studies using OCO-2 data; these observations are best-equipped to constrain the biospheric carbon balance across only continental or
hemispheric regions. By contrast, newer versions of the retrieval algorithm yield a far more detailed constraint, and we are able to constrain $CO_2$ budgets for seven global biome-based regions, particularly during the Northern Hemisphere summer when biospheric $CO_2$ uptake is greatest. Improvements to the OCO-2 observations have had the largest impact on glint mode observations, and we also find the largest improvements in the terrestrial $CO_2$ flux constraint when we include both nadir and
glint data.

## 1 Introduction

Over the past five years, the field of $CO_2$ remote sensing has evolved rapidly. The sheer number of satellites has increased with the launch of TanSat in 2016 (Yang et al., 2018), GOSAT-2 in 2018 (e.g., Masakatsu Nakajima, 2012), and OCO-3 in 2019 (e.g., Eldering et al., 2019). Several additional satellites have also been funded or proposed (e.g., Polonsky et al., 2014;
Tollefson, 2016). In addition, the actual $CO_2$ observations or satellite retrievals have also been changing. Roughly once per year, the NASA Atmospheric $CO_2$ Observations from Space (ACOS) science team releases a new version of the OCO-2 and GOSAT observations that incorporates the most recent advances in the retrieval algorithm and addresses observational errors that have been identified by the scientific community (e.g., O'Dell et al., 2012). Early top-down studies of $CO_2$ fluxes using



OCO-2 employed version 7 of the observations (e.g., Chatterjee et al., 2017; Crowell et al., 2017; Liu et al., 2017; Nassar et al., 2017), but the ACOS team has subsequently updated the observations through version 9 (at the time of writing).

The OCO-2 observations have changed markedly through this process. One of the largest changes occurred with the release of version 8 of the OCO-2 observations in September 2017 (Fig. 1). This update incorporated a multitude of changes to the quality control prescreening process, the forward spectroscopy model, the retrieval algorithm, and the bias correction (O'Dell et al., 2018b). These changes led to widespread improvements in the observations; version 8 has smaller random errors when compared to ground-based observations, a smaller bias between land nadir and land glint observations, and less bias across many northern high-latitude terrestrial regions (Wunch et al., 2017; O'Dell et al., 2018b). Several specific improvements are particularly notable. For example, version 7 had biases greater than 1ppm across the southern ocean that have been remedied in version 8. These errors appeared to be due to high altitude aerosols, so the version 8 algorithm includes a new aerosol layer in the upper troposphere and lower stratosphere that has remedied many of these biases. Furthermore, a correction to the averaging kernel reduced the 0.3ppm bias between land nadir and land glint data (O'Dell et al., 2018b). Overall, the observations rated as good quality in version 8 are very different from those in version 7; 24% of the observations that were marked as high quality in version 7 have been marked as low quality in version 8, and 34% of the observations marked as high quality in version 8 were marked as low quality in version 7.

More recently, version 9 of the OCO-2 observations has been released in October 2018. Improvements in version 9 of the retrieval algorithm yielded smaller changes in the observations (O'Dell et al., 2018a). In particular, this version includes a correction for small-scale biases over land due to topography. Furthermore, the ACOS team relaxed a filter that discards observations collected over dark surfaces, and this change yields more observations over tropical forests (O'Dell et al., 2018a). In spite of these advances, there are still many opportunities for further improving the retrievals. For example, OCO-2 retrievals appear to show biases across most of the northern tropical oceans (O'Dell et al., 2018b).

These improvements to the observations should also improve the robustness of $CO_2$ fluxes estimated using the observations. Several studies indicate that errors in the retrieval can have a substantial impact on strength and robustness of the $CO_2$ flux constraint (e.g., Chevallier et al., 2007; Baker et al., 2010; Crowell et al., 2017; Miller et al., 2018). For example, Miller et al. (2018) explored the robustness of the biospheric $CO_2$ flux constraint using version 7 of the OCO-2 observations. They found that OCO-2 observations can be used to robustly constrain fluxes across continental or hemispheric regions but that the observations have limited ability to robustly estimate biospheric $CO_2$ fluxes across smaller regions. The authors constructed a series of synthetic data experiments to understand the most important factors driving these results; they concluded that atmospheric transport errors and prior flux errors play a role, but retrieval errors are a particularly salient factor. The OCO-2 science team is also developing an ensemble of inverse modeling estimates of $CO_2$ fluxes, and recent comparisons show results that are broadly parallel to Miller et al. (2018): inverse models provide consistent $CO_2$ flux totals for continents or hemispheres but diverge for smaller regions (e.g., Crowell et al., 2017).

The present study is a follow-up to Miller et al. (2018). We re-examine the conclusions of that study in light of recent improvements in OCO-2 observations of $CO_2$. We also identify opportunities for future improvements to the retrievals.





## 2   Methods

### 2.1   Overview

We design a set of top-down experiments to examine whether we can detect variations in biospheric $CO_2$ sources and sinks within different regions of the globe and different months of the year using OCO-2 observations. In the present study, these

variations are defined as any spatial or temporal patterns in $CO_2$ fluxes that have been gridded to the resolution of a global atmospheric model – one degree latitude by one degree longitude and a 3-hourly time interval.

Detecting variations in $CO_2$ fluxes is a pre-requisite for constraining $CO_2$ budgets or flux totals; we must be able to detect variations in $CO_2$ sources and sinks across a region if we are to constrain budgets across any region of smaller size. We begin with large, hemispheric regions and then decrease the size of those regions until we are no longer able to detect any variations

in biospheric $CO_2$ sources and sinks (Fig. 2). That limit is the smallest region for which we could robustly estimate $CO_2$ fluxes using currently-available OCO-2 observations. All of these regions are based on a map of global biomes presented in Olson et al. (2001). The seven-region map contains broad global biomes aggregated from those in Olson et al. (2001) while the two- and four-region maps have been aggregated from Olson et al. (2001) to form even larger regions. We use a biome-based map because inverse modeling studies often estimate $CO_2$ flux totals for biome-based regions, and these regions have clear

ecological significance.

We construct this set of experiments for each of the last three versions of the OCO-2 observations and examine how the results change with the retrieval version. These experiments are identical except for the retrieval version used. Therefore, this setup provides a means to understand how improvements in the observations are improving the constraint on biospheric $CO_2$ fluxes. We examine these questions for each month within the year 2015 – to understand how these results vary by season and

by region or biome.

### 2.2   Implementation of the top-down experiments

We design a regression framework to determine whether we can detect variations in $CO_2$ fluxes using OCO-2 observations. This section provides an overview of the approach, but Miller et al. (2018) provides full descriptive and mathematical detail. This regression will try to match $CO_2$ observations from OCO-2 using numerous atmospheric model outputs. Each model

output estimates the enhancement in total column $CO_2$ ($XCO_2$) from fluxes in a particular region and a particular month. We generate all of these model outputs of $CO_2$ using the Parameterized Chemistry and Transport Model (PCTM) (Kawa et al., 2004). The model setup used here has a spatial resolution of one degree latitude by one degree longitude, and we incorporate $CO_2$ fluxes at a 3-hourly time resolution. The wind fields used to drive PCTM are from the Modern Era Retrospective-Analysis for Research and Applications (MERRA) product (Rienecker et al., 2011). This setup is identical to Miller et al. (2018).

We run many atmospheric model simulations using numerous different biospheric $CO_2$ flux estimates. The regression will try to reproduce OCO-2 observations using a linear combination of these model outputs. For example, in the seven region experiments, we use seven different geographic regions, seven biospheric $CO_2$ flux estimates, and 16 different months (September 2014-December 2015). We discard results from the first four months as model spin-up. These combinations equate to 784 total





atmospheric model outputs. We further run atmospheric model simulations using a spatially and temporally constant flux in each region and each month, and we allow the regression to use these model outputs as well.

This approach provides a means to evaluate when and where current satellite observations can constrain variations in $CO_2$ fluxes. At least some of the former model outputs should help reproduce the OCO-2 observations better than the model outputs

that are driven by spatially and temporally constant fluxes. If so, a model with spatially and temporally variable fluxes is better able to reproduce OCO-2 observations than a model with constant fluxes. This result would imply that OCO-2 observations can be used to detect variations in biospheric $CO_2$ sources and sinks within a given region for a given month. By contrast, suppose that the former model outputs do not reproduce the OCO-2 observations any better than the latter model outputs with constant $CO_2$ fluxes. This result would imply one of several conclusions. First, the observations may not be sensitive to fluxes from

the region or month in question. This outcome may occur if the magnitude of fluxes is small in a given region or if there are no OCO-2 observations near that region. Second, errors in the atmospheric model or in the OCO-2 observations may obscure variations in $XCO_2$ that are due to $CO_2$ fluxes. Lastly, the biospheric $CO_2$ flux estimates used in the atmospheric model may not be skilled and may not reflect real-world biospheric $CO_2$ fluxes. However, in this study, we offer up seven biospheric $CO_2$ flux estimates for each region and each month, and at least one of these estimates should correlate with real-world $CO_2$ fluxes

to a reasonable extent. Hence, it is unlikely that this explanation would drive the results. Rather, it is more likely that the observations are not sensitive to fluxes from a given region or that errors in the model–data system are too large.

Note that we also account for the contribution of non-biospheric fluxes within the regression. Anthropogenic, biomass burning, and ocean fluxes are not the focus of this study. However, we include these fluxes within the regression nonetheless to avoid potentially biasing the results. We model atmospheric enhancements of $XCO_2$ from anthropogenic emissions using

EDGAR v4.2 FT2010 (European Commission, Joint Research Centre (JRC)/Netherlands Environmental Assessment Agency (PBL), 2013; Olivier et al., 2014), climatological ocean fluxes using Takahashi et al. (2016), and biomass burning fluxes using the Global Fire Emissions Database (GFED), version 4.1 (van der Werf et al., 2010; Giglio et al., 2013).

We implement model selection to evaluate when and where current satellite observations can constrain variations in biospheric $CO_2$ fluxes. Model selection will determine which combination of atmospheric model outputs to include in the regres-

sion based upon which best reproduces the OCO-2 observations. If this combination includes at least one biospheric $CO_2$ flux model for a given region and season, we conclude that the observations likely can be used to constrain variations in $CO_2$ fluxes. However, if this combination does not include any biospheric $CO_2$ flux model for a given region and season, we conclude that the observations likely cannot be used to constrain flux variations for that region and season.

We specifically employ a form of model selection known as the Bayesian Information Criterion (BIC), an approach com-

monly used in regression modeling (e.g., Ramsey and Schafer, 2012, chap. 12) and more recently in atmospheric inverse modeling (e.g., Gourdji et al., 2012; Miller et al., 2013; Shiga et al., 2014; Fang et al., 2014; Fang and Michalak, 2015). To this end, we create different combinations of model outputs and use each combination in the regression. We score each combination based upon how well it reproduces the OCO-2 observations; combinations with a lower weighted sum of squares error receive a better score. Each combination is also scored based upon the total number of model outputs in that combination. Specifically,

combinations with a greater number of model outputs receive a larger penalty for complexity, and this penalty prevents com-





binations that overfit the data from receiving an anomalously good score. The best combination of atmospheric model outputs is the one with the lowest score. We subsequently examine this combination and tally whether at least one atmospheric model output using a biospheric flux estimate was selected for each region and each month of the year. Miller et al. (2018) describes this approach in greater detail, including the specific equations for the BIC.

## 3 Results & discussion

### 3.1 Robustness of the biospheric $CO_2$ flux constraint

The constraint on $CO_2$ fluxes using recent versions of the OCO-2 observations is a step-change improvement relative to previous versions. Overall, there was only a limited ability to detect variations in monthly $CO_2$ fluxes across individual biomes using version 7 of the retrievals (Miller et al., 2018, Fig. 3a-c). However, these capabilities have changed using versions 8 and 9 of the observations (Fig. 3d-i). Variations in $CO_2$ fluxes are detectable across tropical biomes much of the year and across temperate biomes in northern hemisphere summer when fluxes from these regions are most variable. These results imply that the updated OCO-2 observations can be used to robustly constrain monthly $CO_2$ fluxes from seven biome-based regions in certain circumstances – in about half of all months in the tropics and during northern hemisphere summer in the extra-tropics.

The improvement in the flux constraint is particularly evident in the four- and seven-region experiments (Figs. 3b-c and 3e-f). In the four-region model selection experiments, the OCO-2 observations provide a robust constraint on tropical fluxes for most months of the year (Fig. 3e). In other words, at least one biosphere flux model is found to explain a sufficiently large fraction of the observed variability in $XCO_2$ as to be selected via the BIC model selection procedure for the tropical regions for most months. This result indicates that spatiotemporal variability in $CO_2$ fluxes from within each of these regions is preserved in the OCO-2 observations. This represents a marked improvement over results when using observations from version 7 of the OCO-2 retrieval algorithm (Figs. 3b and 3e, Miller et al., 2018). The results using the newer versions 8 and 9 also show substantial improvements in other regions, including dryland and dry monsoon regions, temperate regions, and high-latitude regions (Figs. 3e and 3h).

The seven-region model selection experiments are an even more challenging test of current observations. These experiments examine whether we can robustly constraint monthly biospheric fluxes across seven broad, aggregated global biomes. These experiments produce much better results using versions 8 and 9 of the observations. Specifically, biospheric flux models are selected across tropical and subtropical biomes for at least one month of every season. The same is true across all temperate and high-latitude biomes for a minimum of one month during northern hemisphere summer.

One notable feature of all model selection experiments is the result for dryland and dry monsoon regions (Fig. 2c). At first glance, it may appear surprising that biospheric flux models are selected for so many months in this region, given that some parts of this region are very dry and presumably have small $CO_2$ fluxes. Several semiarid regions within this classification have a very distinct monsoon that can bring over 500mm of precipitation per month (e.g., northeastern Brazil, western India, and Pakistan). As a result, there is a large spatial contrast in $CO_2$ fluxes across these regions during northern hemisphere spring



and summer – large $CO_2$ uptake in places with a spring and summer monsoon and little to no fluxes in places like the Sahara or the Arabian Peninsula.

Note that the results using version 9 of the observations are not very different from those using version 8. The change in the observations between versions 8 and 9 is only incremental (e.g., Fig. 1b). Version 9 has a lower quality control threshold
for surfaces with low albedo, resulting in more observations across tropical rainforests (O'Dell et al., 2018a), and this version includes a topography correction that mostly manifests at small spatial scales. The latter change could be very important for studies that estimate point sources or urban emissions using OCO-2. However, these changes are unlikely to make a large difference in this study both given the large size of the regions examined and the $1° \times 1°$ spatial resolution of the atmospheric model simulations.

## 3.2   Drivers of the results

Numerous factors affect the accuracy or robustness of $CO_2$ fluxes estimated from satellite data. These factors include the accuracy of the observations, the atmospheric transport model, and the prior flux estimate used in the inverse model. Arguably, improvements in any of these inverse modeling inputs could improve the constraint on biospheric $CO_2$ fluxes. However, improvements in the observations have arguably been more attainable than these other factors, and we find that these im-
provements are having a large impact on the robustness of the $CO_2$ flux constraint. Furthermore, these improvements are not restricted to a single satellite like OCO-2. Rather, the ACOS retrievals and bias correction will be directly applicable to other NASA carbon monitoring missions, including the recently-launched OCO-3 mission and the planned GeoCarb mission.

These improvements to the retrieval algorithm have had an effect on both glint and nadir observations from OCO-2 collected in almost every region of the globe. The sheer number of different changes makes it challenging to pinpoint exactly which
have had the largest impact on the $CO_2$ flux constraint; there have been numerous updates to the quality control prescreening, the forward spectroscopy model, the retrieval algorithm, and the bias correction. Furthermore, these updates have had multiple effects on the reported $CO_2$ observations, reducing white noise, reducing bias, and changing which observations do or do not pass quality control. O'Dell et al. (2018b) detail these changes in much greater detail.

With that said, a few of these improvements appear to have a particularly salient impact on the results of this study. For
example, the largest improvements have generally been to the glint mode observations. A 0.2 to 0.3 ppm bias between land nadir and land glint observations in version 7 has been remedied in version 8, and version 8 glint observations show smaller biases across many ocean regions. Furthermore, version 8 exhibits less random noise in all types of observations, but that noise reduction is largest in glint observations, both over land and over the oceans (O'Dell et al., 2018b).

Indeed, we also see the largest improvement in the flux experiments conducted in this study when we include glint mode ob-
servations. Figure 4 displays the results of the model selection experiments when the glint data are excluded. The figure shows results using version 7, and 8, and 9 of the observations. The improvement between versions 7 and 8 is much smaller when the glint observations are excluded than when they are included (Fig. 3). Even in terrestrial regions, these glint observations may play a key role in the overall flux constraint. For example, the absolute number of nadir and glint observations over land are





roughly equal; there are $4.3 \times 10^6$ land nadir observations with a positive quality control flag for 2015 and $4.3 \times 10^6$ land glint observations during the same time period.

Note that this study focuses on detecting variations in $CO_2$ fluxes from terrestrial regions in individual months. To that end, certain types of flux estimation problems are beyond the scope of the current study. For example, there is strong evidence that OCO-2 observations are still biased across northern tropical oceans, and reductions in these biases could improve ocean flux estimates derived from OCO-2 (Baker, 2018; O'Dell et al., 2018b). Furthermore, there is always a possibility that the observations have a bias that is correlated across regions larger than those examined in this study. For example, the observations show a small, time-dependent drift from one year to another (O'Dell et al., 2018b). The approach used in this study would be unlikely to detect the impact of those biases.

## 4   Conclusions

$CO_2$ observations from the OCO-2 satellite have changed enormously with recent improvements to the retrieval algorithm. New observations are more self-consistent (e.g., better agreement between glint and nadir data) and compare better against ground-based observations. In some regions, these changes are comparable in magnitude to the atmospheric $CO_2$ enhancement due to biospheric $CO_2$ sources and sinks.

In this study, we specifically examine how these changes to the retrieval algorithm have improved the constraint on biospheric $CO_2$ fluxes, and we find that the improvement is large. Using observations based on version 7 of the retrieval algorithm, we find that biospheric fluxes can only be constrained across continental or hemisphere-size regions, as these observations rarely yield a robust constraint for smaller regions. By contrast, we find a step-change improvement in the biospheric $CO_2$ flux constraint using updated versions of the OCO-2 observations, based on versions 8 and 9 of the retrieval algorithm. Specifically, these improvements make it possible to robustly constrain $CO_2$ fluxes across seven global biome-based regions during many seasons of the year. This improvement is particularly large when both nadir and glint data are included.

This study indicates that improvements to space-based $CO_2$ observations are yielding large improvements in global monitoring of biospheric carbon fluxes. As new $CO_2$ monitoring missions like OCO-3 and GeoCarb launch into orbit, these improvements will have a lasting impact on space-based monitoring of $CO_2$.

*Data availability.* All OCO-2 observations are available from NASA's Goddard Earth Sciences Data and Information Services Center (GES DISC) at https://disc.gsfc.nasa.gov/OCO-2 (last access: 24 Feb. 2019).

*Author contributions.* S.M.M and A.M.M. designed and wrote the study.



*Competing interests.* The authors declare that they have no conflicts of interest.

*Acknowledgements.* We thank Christopher O'Dell, Annmarie Eldering, and David Crisp for their feedback on the research. The OCO-2 data are produced by the OCO-2 project at the Jet Propulsion Laboratory, California Institute of Technology, and obtained from the OCO-2 data archive maintained at the GES DISC. This work is funded by NASA ROSES grant no. 80NSSC18K0976.





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



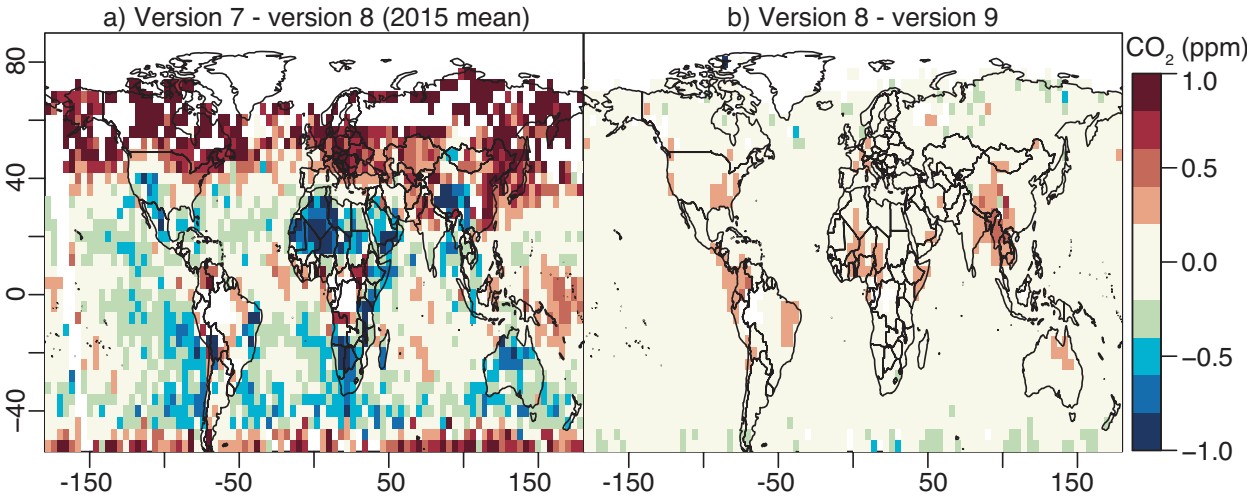

**Figure 1.** Differences between versions 7 and 8 of the OCO-2 observations (a) and between versions 8 and 9 of the observations (b). Version 8 was a much larger update to the observations than version 9. We average all of the differences between observations onto a grid to make the differences more visually apparent. The results shown here are for observations collected in 2015, the time period analyzed in this study. In addition, this map only displays grid boxes with more than 250 total observations in 2015.





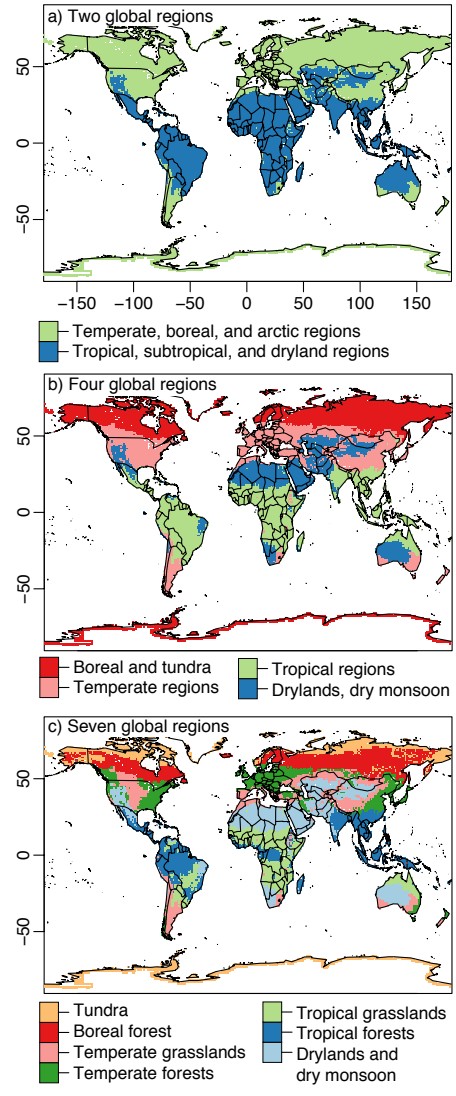

**Figure 2.** The two hemispheric regions (a), four continental regions (b), and seven biome-based regions (c) used in this study. These regions are based upon the world biome map by Olson et al. (2001). The two- and four-region maps are constructed by aggregating individual biomes into larger regions.



**Figure 3.** Results of the model selection experiments using versions 7, 8, and 9 of the OCO-2 observations. Versions 8 and 9 provide a much more robust constraint on biospheric $CO_2$ fluxes than version 7. The top row displays the results of the experiments with two global regions, the second row with four global regions, and the third row with seven global regions. Each box is color-coded based upon the number of months in which at least one biospheric flux model is chosen using model selection. Dark colors indicate a robust constraint on monthly $CO_2$ fluxes while light colors indicate a weak constraint. Note that these experiments include nadir, target, glint mode observations. In addition, version 7 results are the same as those in Miller et al. (2018).





**Figure 4.** Results of the model selection experiments using only nadir and target mode observations. The improvement between versions 7 and 8 is less pronounced when we exclude glint observations and include only nadir and target mode data. Version 7 results here are the same as those in Miller et al. (2018).