# Peer review of "The impact of improved satellite retrievals on estimates of biospheric carbon balance"

_Atmospheric Chemistry and Physics, 2019_

## Referee Comment (RC1) · Anonymous Referee #1 · 18 Jun 2019

This paper is an extension of a previous study that documented the information content of the OCO-2 retrievals in their version 7 (Miller et al., 2018). The extension concerns the improvement brought by versions 8 and 9. It could have been anecdotal but the results are striking enough to warrant publication, in particular given the wide use of version 7 (e.g., Crowell et al., 2019). It could also contribute to explain the increased realism claimed by some inversion results with version 9 (Chevallier et al., 2019, http://dx.doi.org/10.5194/acp-2019-213). The paper is concise, well written and quite pedagogical. I recommend publication after a few issues can be addressed.

- Throughout the text, the authors use the expression "robust constraint", but what is it? If for instance all OCO-2 L4 products had no better quality than the latest biosphere models at any scale, it could be found useless for land vegetation

carbon accounting and therefore not robust for that application. I do not think that the chosen method can conclude to robustness. The authors need to qualify their conclusion better: they demonstrate improvement in the retrievals on the basis of a specific indicator, but what does this mean in practice?

- Crowell et al. (2017) should be updated to Crowell et al. (2019, http://dx.doi.org/10.5194/acp-2019-87).

- P. 3, l.9: the authors actually do not use more than 7 biome regions and therefore do not necessarily reach the point when they are no longer able to detect any variations in biospheric CO2 sources and sinks.

- P. 3, l. 19: the choice of a year with a strong El Nino episode is surprising. How would the results change with a "normal" year?

- P. 3, l. 32: the authors need to give details about the seven models so that the reader can get convinced about their realism. For instance, I understand that Miller et al. (2018) used climatological model averages for technical reasons (lack of model availability for the target year): now that model outputs for 2015 are widely available, has this issue been sorted out?

- P. 6, l. 14: I have not seen that the community has deployed significant effort to improve their transport models or their error models in the past years. In comparison, the effort on retrievals, in particular in the OCO-2 team, has been huge. It is not fair to compare them to the rest.

- Legends of Figs. 3 and 4: what are target mode retrievals doing here?

---

## Referee Comment (RC2) · Anonymous Referee #2 · 7 Aug 2019

This is a well-written study and concise analysis of the improved constraint on biospheric CO2 fluxes associated with improvements to the retrieval algorithm for OCO-2. Some major and minor comments on the manuscript are noted below.

Major comments:

* It is interesting that the retrieval bias reductions from Version 7 to 8 helped so much with the biospheric flux constraint at the biome-scale. It would be nice for the authors to comment a little more on subtle differences between versions 8 and 9. Looks like the constraint went down in some regions, e.g. the drylands and dry monsoon areas. Why is that?

* Did you try estimating any sub-biome scale regions? Given that the biomes tend to

be multi-continental, it would be interesting to see the results using smaller regions that are (mostly) spatially contiguous within a given continent, especially with Versions 8 & 9. The statement on p. 3, lines 8-10 sounds somewhat misleading: "We begin with large, hemispheric regions and then decrease the size of those regions until we are no longer able to detect any variations in biospheric CO2 sources and sinks." It looks like you could potentially go to even finer spatial scales in the tropical grasslands/ forests and drylands/ dry monsoon biomes with Version 8 & 9 retrievals.

* This may not be the focus of your study, but I was very curious to see the results of your model selection and estimated betas from the regression with the selected bio models (and anthro/ biomass burning/ ocean fluxes). Which biospheric models were selected in different region/ month combinations? When was just one model selected vs. multiple models? Can these results help to inform which models are performing best in which regions? Does the "best" model for a given month change as a function of spatial scale? This could be potentially useful information for biospheric model developers. Also, I don't see a supplemental material, but do you list anywhere which bio models went into the model selection algorithm?

* Not clear why you would include or exclude glint observations. It looks like in Miller et al, 2018, you exclude glint observations from results shown in the main manuscript. Why? How has the quality of these observations improved in Versions 8 and 9? And why are glint observations helping especially in tropical regions? Are they able to improve the density of observations in cloud-covered areas, or is a single glint measurement more informative than a single nadir or target measurement in these regions? Please don't assume too much satellite-based knowledge on the part of the reader!

Minor comments:

* P. 3, lines 31-33: it might be nice to put an equation or diagram or even table here showing the potential inputs that go into the model selection and your regressions. Do you run model selection on all months simultaneously? That's what it sounds like, but

please make that more clear.

* P. 4, lines 4 and 8: please replace the terms "former" and "latter" with something more descriptive, e.g. biospheric model output and constant fluxes.

* P. 4, line 19: "to avoid potentially biasing the results". This is true, but please make clear that XCO2 reflects the contributions of all these different types of fluxes (ocean/ FF/ BB/ terrestrial bio), so you need to account for the non-bio fluxes in order to isolate the signal of the bio in the regression. Can also comment that the uncertainty on the FF/ ocean/ BB fluxes is thought to be much smaller than that on the terrestrial bio fluxes (with reference).

* P. 5, line 13: "in about half of all months in the tropics", but didn't you say on line 10 that "variations in CO2 fluxes are detectable across tropical biomes much of the year?" In Version 9, it looks like you can constrain bio fluxes in the tropical grasslands and forests for 8 and 9 months of the year, respectively.

* P. 6, line 17: please add references for the ACOS retrievals and bias correction, and also for OCO-3 and GeoCarb.

* P. 7, lines 6 to 9: it is not clear to me, at least, why a bias correlated across regions larger than those examined in this study (e.g. a time-dependent drift for the whole globe) could potentially qualitatively impact the results of your study. Given that you are looking at a single year, would this time-dependent drift matter at all? Wouldn't the constant component in your regression account for this global bias between obs and convolutions?

---

## Author Response (AR1)

We thank the referees for their comments and suggestions on the manuscript. Below, we have included the referee's point-by-point suggestions and the associated changes we have made to the manuscript.

- "Throughout the text, the authors use the expression "robust constraint", but what is it? If for instance all OCO-2 L4 products had no better quality than the latest biosphere models at any scale, it could be found useless for land vegetation carbon accounting and therefore not robust for that application. I do not think that the chosen method can conclude to robustness. The authors need to qualify their conclusion better: they demonstrate improvement in the retrievals on the basis of

a specific indicator, but what does this mean in practice?"

We have removed the word "robust" throughout the text when it is used to refer to the flux constraint. The methods section of the article describes, in detail, how the experiments are set up and what they do and do not indicate about the $CO_2$ flux constraint. Elsewhere in the article, we often use shorter, more concise language to refer to these experiments. Where possible, we have tried to use more specific wording throughout the entire article. Specifically, we have replaced the word "robust" in the following instances throughout the text:

Pg. 2, line 22: replaced with "reliability or accuracy"

Pg. 2, line 23: deleted "and robustness"

Pg. 2, line 25: replaced "robustness" with "detectability"

Pg. 2, line 26: replaced "can be used to robustly constrain fluxes across" with "can be used to identify variations in biospheric fluxes within"

Pg. 2, line 27: deleted "robustly"

Pg. 3, line 10: We have removed this sentence in response to another review comment.

Pg. 5, line 6: replaced "robustness" with "strength"

Pg. 5, line 12: replaced "robustly constrain" with "detect and constrain variations in"

Pg. 5, line 15: replaced "provide a robust constraint" with "can be used to detect variations in"

Pg. 5, line 24: replaced "robustly constraint monthly biospheric fluxes" with "detect spatiotemporal variations in biospheric fluxes"

Pg. 6, line 11: deleted "or robustness"

Pg. 6, line 15: We have edited this sentence in response to another reviewer suggestion.

Pg. 7, line 18: replaced "as these observations rarely yield a robust constraint for smaller regions" with "as these observations can rarely be used to detect or constrain variations in $CO_2$ fluxes across smaller regions".

Fig. 3 caption: deplaced "more robust" with "stronger"

- "Crowell et al. (2017) should be updated to Crowell et al. (2019, http://dx.doi.org/10.5194/acp-2019-87)"

We have updated this reference in the revised manuscript.

- "P. 3, l.9: the authors actually do not use more than 7 biome regions and therefore do not necessarily reach the point when they are no longer able to detect any variations in biospheric $CO_2$ sources and sinks."

We have clarified the text here. We use very large regions in the first two sets of experiments and then shrink those regions down to biome-sized regions in the final set of experiments. This final set of experiments is both a challenging test of current observations and would be an ambitious, ecologically-relevant goal for future inverse modeling studies.

- "P. 3, l. 19: the choice of a year with a strong El Nino episode is surprising. How would the results change with a "normal" year?"

We began working on the preceding companion paper in 2016, and at that time, there was only a single year of OCO-2 observations available to analyze. Hence, both that paper and the current manuscript focus on OCO-2 observations from 2015. In the current manuscript, we have examined the same time period as in the preceding companion paper – to ensure that we can make an apples-to-apples comparison between the two studies. We suspect that results for 2016 would be similar to the analysis for 2015. Environmental conditions in some regions were different in 2015 relative to 2016 due to El Nino, but those differing conditions should not interfere with the regression analysis used in this study;

many of the predictor variables used in the analysis would differ in 2015 and 2016 to reflect these differing environmental conditions (e.g., EVI, NDVI, and SIF).

- "P. 3, l. 32: the authors need to give details about the seven models so that the reader can get convinced about their realism. For instance, I understand that Miller et al. (2018) used climatological model averages for technical reasons (lack of model availability for the target year): now that model outputs for 2015 are widely available, has this issue been sorted out?"

We have added an SI to the manuscript that describes each of these seven models. This information is also described in the preceding companion paper, and the information in this SI is a duplicate of the information in the preceding companion paper.

Model outputs for 2015 were not available at the time that we began work on the preceding companion paper, and we want to compare apples-to-apples with that paper. There are now biospheric model outputs available for 2015. However, we require a relatively large number of flux model estimates for the statistical model, and there are not a sufficient number of biospheric model outputs that are readily available at a 3-hourly time resolution for 2015. The creation of a new flux model inter-comparison was beyond the scope of the current project. With that said, we have incorporated numerous vegetation indices for 2015 within the statistical model, including SIF, EVI, and NDVI.

- "P. 6, l. 14: I have not seen that the community has deployed significant effort to improve their transport models or their error models in the past years. In comparison, the effort on retrievals, in particular in the OCO-2 team, has been huge. It is not fair to compare them to the rest."

We have clarified this statement in the revised version of the manuscript, and we have deleted the phrase about retrieval improvements being more attainable than improvements in transport modeling. Our intent here is not to compare improve-

ments in the retrievals against improvements in meteorology or in biospheric flux modeling. Rather, we wanted to point out that the retrievals, while important, are one factor among many that affect the $CO_2$ flux constraint.

- "Legends of Figs. 3 and 4: what are target mode retrievals doing here?"

  We did not see any reason to exclude target mode observations from the analysis. For example, O'Dell et al. (2018) describe the version 8 ACOS retrieval, and they do not present any evidence to indicate anomalous errors or biases in the target mode observations. We also included target mode observations in the analysis in the preceding companion manuscript, and we want to compare apples-to-apples with the results of that study. The objective of the present manuscript is to compare how the flux constraint has improved as the retrievals have evolved from version 7 to versions 8 and 9. We feel it would be difficult to make that comparison if we used a different approach to analyze versions 8 and 9 than we used to analyze version 7 in the preceding manuscript.

**References**

O'Dell, C. W., Eldering, A., Wennberg, P. O., Crisp, D., Gunson, M. R., Fisher, B., Frankenberg, C., Kiel, M., Lindqvist, H., Mandrake, L., Merrelli, A., Natraj, V., Nelson, R. R., Osterman, G. B., Payne, V. H., Taylor, T. E., Wunch, D., Drouin, B. J., Oyafuso, F., Chang, A., McDuffie, J., Smyth, M., Baker, D. F., Basu, S., Chevallier, F., Crowell, S. M. R., Feng, L., Palmer, P. I., Dubey, M., Garcìa, O. E., Griffith, D. W. T., Hase, F., Iraci, L. T., Kivi, R., Morino, I., Notholt, J., Ohyama, H., Petri, C., Roehl, C. M., Sha, M. K., Strong, K., Sussmann, 15 R., Te, Y., Uchino, O., and Velazco, V. A.: Improved retrievals of carbon dioxide from Orbiting Carbon Observatory-2 with the version 8 ACOS algorithm, Atmos. Meas. Tech., 11, 6539-6576, https://doi.org/10.5194/amt-11-6539-2018, 2018.

Atmos. Chem. Phys. Discuss.,
https://doi.org/10.5194/acp-2019-382-AC2, 2019

[Figure]

We thank the referees for their comments and suggestions on the manuscript. Below, we have included the referee's point-by-point suggestions and the associated changes we have made to the manuscript.

- "It is interesting that the retrieval bias reductions from Version 7 to 8 helped so much with the biospheric flux constraint at the biome-scale. It would be nice for the authors to comment a little more on subtle differences between versions 8 and 9. Looks like the constraint went down in some regions, e.g. the drylands and dry monsoon areas. Why is that?"

  We have added text to the revised manuscript to clarify these differences. These

  small differences are due to the stochastic nature of the statistical model. The regression model used in this manuscript requires an estimate of error variances and estimates of the error correlation length and correlation time. We estimate these variances and covariances using a randomized sub-selection of the observations, described in the preceding companion paper; there are too many OCO-2 observations over a year to use all of the observations in that estimation process. Hence, the results of the regression analysis exhibit a small amount of stochasticity depending upon precisely which observations were randomly selected for the variance and covariance estimation. For example, for the simulations shown in the manuscript, we obtained a slightly higher error variance for version 9 $((0.90 \text{ ppm})^2)$ than version 8 $((0.87)^2 \text{ ppm}^2)$ and a slightly longer decorrelation length. This resulted in model selection results for version 9 in which slightly fewer months were selected relative to version 8. We subsequently re-ran the analysis and then obtained a slightly lower error variance for version 9 relative to version 8 $((0.83 \text{ ppm})^2 \text{ versus } (0.87)^2 \text{ ppm}^2)$. This resulted in model selection results for version 9 in which slightly more months were selected relative to version 8. We have added a brief description of this point in the revised manuscript.

- "Did you try estimating any sub-biome scale regions? Given that the biomes tend to be multi-continental, it would be interesting to see the results using smaller regions that are (mostly) spatially contiguous within a given continent, especially with Versions 8  9."

  It could be interesting to examine sub-biome scale regions. However, the overall motivation of this study was to compare apples-to-apples with the preceding companion paper. In that study, we did not examine smaller regions because we had limited success in constraining fluxes across biome-sized regions. In the present manuscript, by contrast, we were able to detect spatiotemporal variations in $CO_2$ fluxes within many of these biome-sized regions, a large improvement over results using version 7 of the observations.

- "The statement on p. 3, lines 8-10 sounds somewhat misleading: 'We begin with large, hemispheric regions and then decrease the size of those regions until we are no longer able to detect any variations in biospheric $CO_2$ sources and sinks.' It looks like you could potentially go to even finer spatial scales in the tropical grasslands/ forests and drylands/ dry monsoon biomes with Version 8 9 retrievals."

The reviewer raises a good point, and reviewer #1 made a similar suggestion. We have revised this statement in the manuscript accordingly.

- "This may not be the focus of your study, but I was very curious to see the results of your model selection and estimated betas from the regression with the selected bio models (and anthro/ biomass burning/ ocean fluxes). Which biospheric models were selected in different region/ month combinations? When was just one model selected vs. multiple models? Can these results help to inform which models are performing best in which regions? Does the 'best' model for a given month change as a function of spatial scale? This could be potentially useful information for biospheric model developers. Also, I don't see a supplemental material, but do you list anywhere which bio models went into the model selection algorithm?"

We agree; model selection can be a useful tool to help identify patterns in $CO_2$ fluxes that are or are not consistent with atmospheric observations. A number of studies have used model selection to explore which flux patterns and which biosphere models are best able to reproduce atmospheric observations. For example, Fang et al. (2014) and Fang and Michalak (2015) explore these questions using in situ $CO_2$ observations. We agree that these are interesting questions but feel that these questions are beyond the scope of the current study and would be better answered in a separate future, study. Adding that analysis to the present

study would arguably complicate or distract from the framing and messaging of the current manuscript.

- "Not clear why you would include or exclude glint observations. It looks like in Miller et al, 2018, you exclude glint observations from results shown in the main manuscript. Why? How has the quality of these observations improved in Versions 8 and 9? And why are glint observations helping especially in tropical regions? Are they able to improve the density of observations in cloud-covered areas, or is a single glint measurement more informative than a single nadir or target measurement in these regions? Please don't assume too much satellite-based knowledge on the part of the reader!"

The reviewer makes a really good point about not assuming too much satellite-based knowledge on the part of the reader. We have added more explanation on this topic in the revised manuscript. In brief, glint observations have historically had much higher error variances and larger biases relative to nadir observations. For example, land glint observations in version 7 had a $\sim$0.5ppm offset compared to land nadir observations (e.g., O'Dell et al. 2018). Until recently, it was arguably very challenging to include both types of observations in an inverse model because one type had a fundamentally different magnitude relative to the other. In the preceding companion paper (Miller et al. 2018), we included results using glint observations within the SI, but we did not put great emphasis on these results with glint observations because of their known biases.

By contrast, the accuracy of the glint observations greatly improved markedly with version 8 of the observations. In fact, the largest improvements between versions 7 and 8 of the observations was to the glint observations, and these improvements greatly reduced the bias between land nadir and land glint observations (O'Dell et al. 2018). These improvements arguably make it feasible to assimilate land nadir and land glint observations in the same top-down framework or inverse model.

We have also included more explanation in the revised manuscript about the improvements tropical biomes versus mid- and high-latitude biomes. The results using versions 7 and 8 show the greatest differences across tropical biomes. This feature is most likely because there is a large signal-to-noise ratio in many tropical biomes throughout the year, whereas the signal-to-noise ratio in mid- and high-latitudes is only large during northern hemisphere summer. Phrased differently, there is a consistent flux signal from many tropical regions throughout the year, and hence we are able to detect variations in fluxes from tropical regions across different seasons using version 8 of the observations. By contrast, net ecosystem exchange (NEE) in northern mid- and high-latitudes has the largest absolute magnitude during northern hemisphere summer. As a result, we see a large improvement in the flux constraint in mid-latitudes in northern hemisphere summer but not in other times of year when the absolute magnitude of NEE is smaller. Furthermore, there are far fewer land nadir and land glint observations in northern mid- and high-latitudes in northern hemisphere winter.

- "* P. 3, lines 31-33: it might be nice to put an equation or diagram or even table here showing the potential inputs that go into the model selection and your regressions. Do you run model selection on all months simultaneously? That's what it sounds like, but please make that more clear."

We have added text to the revised manuscript to clarify. We do run all months simultaneously. We have also added an equation to the manuscript to summarize the regression:

$z = h(\mathbf{X})\beta + b + \epsilon$

where $z$ are the OCO-2 observations, $\mathbf{X}$ the different predictor variables, h() an atmospheric transport model (in this case PCTM), $\beta$ the coefficients estimated in the regression, $b$ the model spinup or $CO_2$ mixing ratios at the beginning of the experiments, and $\epsilon$ the model–data residuals. Note that there are different columns of $\mathbf{X}$ corresponding to each biospheric flux model in each different month and

each different biome. Model selection will determine which columns of $\mathbf{X}$ can best reproduce the OCO-2 observations without overfitting those observations.

This equation and the associated explanation is also included in the preceding companion paper.

- "* P. 4, lines 4 and 8: please replace the terms 'former' and 'latter' with something more descriptive, e.g. biospheric model output and constant fluxes."

We have edited the text accordingly. We have replaced the word "former" with "some of the model outputs that use a flux model or vegetation index," and we have removed the word "latter."

- "* P. 4, line 19: 'to avoid potentially biasing the results'. This is true, but please make clear that XCO2 reflects the contributions of all these different types of fluxes (ocean/ FF/ BB/ terrestrial bio), so you need to account for the non-bio fluxes in order to isolate the signal of the bio in the regression. Can also comment that the uncertainty on the FF/ ocean/ BB fluxes is thought to be much smaller than that on the terrestrial bio fluxes (with reference)."

The reviewer makes a great point, and we have edited the text accordingly.

We have also added text to the manuscript explaining that biospheric fluxes are thought to be more uncertain than other $CO_2$ source types. For example, we have cited the National Academy of Science Report on fossil fuel $CO_2$ emissions (NAS 2010) and have cited a biosphere flux model intercomparison paper (Huntzinger et al. 2012) and a Global Carbon Project assessment (Le Quéré et al. 2018) as evidence of these differing uncertainties.

- "* P. 5, line 13: 'in about half of all months in the tropics', but didn't you say on line 10 that 'variations in $CO_2$ fluxes are detectable across tropical biomes much of the year?' In Version 9, it looks like you can constrain bio fluxes in the tropical grasslands and forests for 8 and 9 months of the year, respectively"

The reviewer is correct – the number cited in the manuscript should be two thirds, not one half. That is an error on our part. We have updated the text accordingly.

- "P. 6, line 17: please add references for the ACOS retrievals and bias correction, and also for OCO-3 and GeoCarb."

We have added references to this line accordingly. We have added citations to O'Dell et al. (2012) and O'Dell et al. (2018) for the ACOS retrieval, Eldering et al. (2019) for OCO-3, and Polonsky et al. (2014) for GEOCarb.


These experiments are based upon a regression framework, as described in the main article. The regression has the following form:

$$z = h(\mathbf{X})\boldsymbol{\beta} + \boldsymbol{b} + \boldsymbol{\epsilon} \tag{S1}$$

where $z$ (dimensions $n \times 1$) are the OCO-2 observations, $\mathbf{X}$ ($m \times p$) contains $p$ different $CO_2$ flux tracers. These tracers include terrestrial biosphere model (TBM) estimates of $CO_2$ fluxes and remote sensing vegetation indices that are known to correlate with patterns in $CO_2$ fluxes (Sect. S2). There are different columns of $\mathbf{X}$ corresponding to each $CO_2$ flux tracer in each different month and each different global region; we run the regression on all months simultaneously. These tracers (both the TBMs and vegetation indices) are subsequently run through an atmospheric transport model $h()$, in this case the Parameterized Chemistry and Transport (PCTM) model (Kawa et al., 2004). The coefficients estimated as part of the regression ($\boldsymbol{b}$, $n \times 1$) scale these model outputs to best match the observations ($z$). Furthermore, $\boldsymbol{b}$ ($n \times 1$) is the model spinup or $CO_2$ mixing ratios at the beginning of the experiments, and $\boldsymbol{\epsilon}$ ($n \times 1$) are the model–data residuals.

We pair this regression with model selection; model selection will determine which combination of model outputs (i.e., columns of $h(\mathbf{X})$) best describe variability in current OCO-2 observations. It will identify the set of model outputs with the greatest power to describe the data and ensures that the regression does not overfit the data (e.g., Zucchini, 2000). We specifically implement model selection based on the Bayesian Information Criterion (BIC), one of the most commonly-used forms of model selection (Schwarz, 1978; Mueller et al., 2010; Gourdji et al., 2012). We calculate a BIC score for many different combinations of model outputs, and each combination has a different set of columns ($h(\mathbf{X})$). The best combination has the lowest BIC score:

$$BIC = L + p \ln(n^*) \tag{S2}$$

where $L$ is the log likelihood of a specific combination of model outputs, $p$ is the number of model outputs in that combination, and $n^*$ is the effective number of independent observations from OCO-2 during the study period. The log likelihood equation rewards combinations of model outputs that improve model-data fit, described in detail in Miller et al. (2018). By contrast, the second term in the equation ($p \ln n^*$) penalizes combinations with a greater number of model outputs, and it ensures that the selected combination is not an over-fit. This penalty and the log likelihood ($L$) not only depend upon the number of model outputs but also the effective number of independent observations ($n^*$). This number reflects the level of spatial and temporal correlation in the observational and model errors. If the spatial and temporal error correlations are large (i.e., bias-type errors), then the $n^*$ will be small relative to the total number of OCO-2 observations. By contrast, if the errors are uncorrelated and completely independent, then $n^*$ will equal the total number of OCO-2 observations. The companion paper Miller et al. (2018) describes in detail how we estimate this quantity.

**S2   Additional detail on the tracers used in model selection**

This section provides additional detail on the TBMs and vegetation indices that are used in the model selection experiments. These TBMs and vegetation indices are used as the input tracers in PCTM. We then generate forward atmospheric model simulations using PCTM and interpolate these model outputs to the locations and times of the OCO-2 observations to generate modeled $XCO_2$ total columns. These modeled columns become the columns of $h(\mathbf{X})$ in the model selection experiments (Eq. S1). Note that the multiple regression will scale the magnitude of the TBMs and vegetation indices in each region and each month to best match the observations (Eq. S1). As a result of this setup, the overall magnitude of each TBM and of each vegetation index does not affect the model selection results. Rather, this study assesses the degree to which the spatial and temporal patterns in the TBMs and vegetation indices, after being transported through the atmosphere to the times and location of OCO-2 observations, can explain the spatial and temporal patterns in OCO-2 observations.

We include four TBMs from the recent MsTMIP project (Huntzinger et al., 2013). These TBMs have very different space-time patterns and therefore represent a wide range of plausible flux patterns. The TBMs include the Dynamic Land Ecosystem Model (DLEM; e.g., Tian et al., 2011) , the Lund-Potsdam-Jena Model Wald Schnee und Landschaft version (LPJ; e.g., Sitch et al., 2003) , the Global Terrestrial Ecosystem Carbon Model (GTEC; e.g., King et al., 1997), and the Simple Biosphere Model with the Carnegie-Ames-Stanford Approach (SIBCASA; e.g., Schaefer et al., 2008) . The original MsTMIP products have a spatial resolution of 0.5° latitude by 0.5° longitude, and we regrid these products to the PCTM model grid (1° latitude by 1.25° longitude). Furthermore, Fisher et al. (2016) downscaled the MsTMIP products to a 3-hourly temporal resolution; we use this version of the MsTMIP products in the present study.

The MsTMIP estimates are available through year 2010. Because these estimates are not available for the years of this study (2014-2015), we use a multi-year average as inputs in the PCTM model. Specifically, downscaled MsTMIP products are available from Fisher et al. (2016) for years 2004-2010, and we average the MsTMIP models over those years within each separate model grid box and for each separate 3-hourly time period to produce a multi-year average for each MsTMIP estimate. The resulting $CO_2$ flux estimates vary hour to hour and day to day but not year to year. Note that some recent inverse modeling studies using OCO-2 observations incorporate a prior flux estimate that has been generated for more recent years (e.g., Crowell et al., 2019). Unlike inverse modeling studies that often require a single prior flux estimate, we require numerous 3-hourly $CO_2$ flux tracers that represent a wide range of plausible patterns for the statistical model used in this study. The creation of a new, updated TBM inter-comparison is beyond the scope of this study. Furthermore, the objective of this study is to compare how the $CO_2$ flux constraint has improved as the retrievals have evolved from version 7 to versions 8 and 9. To facilitate this comparison, we have used the same set of flux models from MsTMIP as in the preceding companion paper (Miller et al., 2018).

In addition to these TBMs, we also utilize several vegetation indices as possible tracers of $CO_2$ fluxes within the regression ($\mathbf{X}$ in Eq. S1). These include the enhanced vegetation index (EVI), normalized difference vegetation index (NDVI), and solar-induced fluorescence (SIF). Numerous studies indicate that biospheric $CO_2$ fluxes correlate with these vegetation indices – with EVI (e.g., Sims et al., 2008; Wu et al., 2011), NDVI (e.g., Cihlar et al., 1992; Wylie et al., 2003) , and SIF (e.g., Guanter et al., 2014; Yang et al., 2015; Shiga et al., 2018). These indices are therefore good candidate $CO_2$ flux tracers to use within the model selection experiments.

We specifically use EVI and NDVI estimates from the Moderate Resolution Imaging Spectroradiometer (MODIS) Aqua product MYD13C1 (Didan, 2015a) and the MODIS Terra product MOD13C1 (Didan, 2015b). These products are collectively available at 8-day intervals. The individual Aqua and Terra products are each available at 16-day intervals. However, the two products are staggered, so Aqua and Terra can be combined to produce EVI and NDVI estimates every

8 days. These products have a 0.05° latitude by 0.05° longitude, and we regrid them to the PCTM model grid (1° latitude by 1.25° longitude). Both of these products are available for 2014 and 2015, the time period of this study.

We also use level 2 SIF retrievals from the Global Ozone Monitoring Experiment-2 (GOME-2) (Joiner, 2014). We convert the level 2 retrievals to a gridded SIF product using a block kriging method described by Tadić et al. (2017). This gridded product has a daily temporal resolution and the same spatial resolution as PCTM. We use this product an input to the PCTM model and incorporate the resulting model outputs as candidate variables in the $h(\mathbf{X})$ matrix.

**S3  Differences in the model selection results for versions 8 and 9 of the observations**

The model selection results using versions 8 and 9 of the observations are very similar but exhibit a few subtle differences (Fig. 3 and 4). Specifically, we select slightly fewer $CO_2$ flux tracers using version 9 than version 8 in Fig. 3. These small differences are due to the stochastic nature of the statistical model. The regression model used in this manuscript requires an estimate of error variances and estimates of the error correlation length and correlation time. These estimates are used to calculate $n^*$ (Eq. S2). We estimate these parameters using a randomized sub-selection of the observations due to the very large size of the OCO-2 dataset, a procedure described in Miller et al. (2018). The results of the regression analysis therefore exhibit a subtle stochasticity depending upon which observations were randomly selected for the variance and covariance estimation. For example, when we re-run the analysis in Fig. 3, we sometimes select a flux model in one or two more months using version 9 relative to version 8 and sometimes obtain identical results using versions 8 and 9.